# The Protective Effect of Bariatric Surgery on the Development of Colorectal Cancer: A Systematic Review and Meta-Analysis

**DOI:** 10.3390/ijerph20053981

**Published:** 2023-02-23

**Authors:** Nikolaos Pararas, Anastasia Pikouli, Dionysios Dellaportas, Constantinos Nastos, Anestis Charalampopoulos, Mohamad Ayham Muqresh, George Bagias, Emmanouil Pikoulis, Dimitrios Papaconstantinou

**Affiliations:** 1Third Department of Surgery, Attikon University Hospital, National and Kapodistrian University of Athens, 15772 Athens, Greece; 2College of Medicine, Sulaiman Al Rajhi University, Al-Qassim 52726, Saudi Arabia

**Keywords:** colorectal cancer, obesity, bariatric surgery, meta-analysis

## Abstract

Background: Obesity is a known risk factor for developing colorectal cancer (CRC) and is associated with the formation of precancerous colonic adenomas. Bariatric surgery (BRS) is considered to reduce the cancer risk in morbidly obese patients. However, the currently available literature yields contradicting results regarding the impact of bariatric surgery on the incidence of CRC. Methods: A systematic literature search of the Medline, Embase, CENTRAL, CINAHL, Web of Science, and clinicaltrials.gov databases was undertaken following the PRISMA guidelines. A random effects model was selected. Results: Twelve retrospective cohort studies, incorporating a total of 6,279,722 patients, were eligible for inclusion in the final quantitative analysis. Eight studies originated from North America, while four reported on European patients. Patients in the Bariatric Surgery group exhibited a significantly reduced risk for developing colorectal cancer (RR 0.56, 95% CI 0.4–0.8, *p* < 0.001), while sleeve gastrectomy was found to be significantly associated with a smaller incidence of CRC (RR 0.55, 95% CI 0.36–0.83, *p* < 0.001), and gastric bypass and banding did not. Conclusions: A significant protective effect of BRS against the development of CRC is implied. In the present analysis, the incidence rate of colorectal cancer was approximately halved amongst the obese individuals that were operated on.

## 1. Introduction

Obesity is a serious worldwide concern with significant implications for Global Public Health. In a 2016 report, the World Health Organization estimated that more than 1.9 billion adults (18 years old and older) worldwide were overweight, with 650 million of them being obese. Overall, about 13% of the world’s adult population (11% of men and 15% of women) is obese, with the prevalence having nearly tripled between the years 1975 and 2016 [1,2]. Concurrently, CRC is responsible for 9.7% of all cancer cases and 8,5% of all the deaths related to cancer worldwide, and its burden will have globally increased by 60% by the year 2030 [3].

It is a well-established fact that the risk of developing several types of cancers is increased in the morbidly obese population. The International Agency for Research on Cancer (IARC) has designated 14 types of cancer as obesity-related ones, including colorectal cancer (CRC) [4,5,6,7]. Obesity alone is postulated to be responsible for 20% of all cancer cases, with a special predilection towards females [6]. The most effective treatment of morbid obesity is thought to be bariatric surgery, which is able to provide sustained, long-term weight loss and improvement or complete remission of the obesity-related comorbidities [8]. Bariatric surgery (BRS) has expanded its applications in the past three decades, and it is, at present, among the most frequently performed gastrointestinal operations worldwide. However, the existing studies on how BRS affects CRC risk often present conflicting results [9,10].

This meta-analysis aims to elucidate the effect of bariatric surgery on CRC by investigating the incidence rates of colorectal neoplasias in obese patients that have undergone a bariatric operation versus those who have not and to evaluate the impact of each bariatric procedure on the incidence of CRC.

## 2. Methods

### 2.1. Literature Search and Study Selection Process

A systematic literature search of the Medline, Embase, CENTRAL, CINAHL, Web of Science, and clinicaltrials.gov databases was undertaken using the search terms “colorectal cancer”, “colorectal neoplasms”, colorectal adenoma”, “colorectal polyp”, “obesity”, “overweight”, and “bariatric”, combined with the Boolean operators AND/OR as appropriate for each database. The search for Medline was phrased as follows: colorectal cancer AND (obese OR obesity OR overweight) AND (bariatric). After the removal of duplicate studies, the generated abstract list was independently screened by two authors (AP and MM), for studies reporting the comparative incidences of colorectal cancer in patients with a history of bariatric surgery (BRS group) and those without (control group). The reference lists of potentially relevant studies were screened for further eligible studies.

All of the studies, irrespective of language, deemed to be relevant by the screening process were evaluated in full independently by two authors (AP and MM), with a third one (DP) acting as a referee in cases of disagreement. The predetermined set of exclusion criteria were: (1) case reports and non-clinical studies, (2) studies reporting neoplasms other than those of colorectal origin, (3) studies without a comparator control group, (4) studies exhibiting population overlap or originating from the same database (the study with the largest sample size was selected), (5) studies not reporting colorectal cancer incidence rates, and (6) studies focusing on specific population subsets (i.e., pediatric).

The present systematic review and meta-analysis was conducted according to PRISMA guidelines [11] and was registered in the International Prospective Register for Systematic Reviews (PROSPERO ID: CRD42023387393).

### 2.2. Data Extraction and Outcomes of Interest

Data extraction was performed by two authors (AC and GB) to ensure data accuracy and completeness. The primary variables of interest were the incidence rates of colorectal cancer and colorectal adenomas or polyps and the type of bariatric surgery that had previously been performed. The secondary outcomes of interest were the patient demographics, the interval of data collection, the country of origin, and the type of database reported. The extracted data were entered into standardized Excel spreadsheets (Microsoft, Redmond, WA, USA) for further tabulation.

### 2.3. Risk of Bias Assessment

The risk of bias was evaluated independently by two authors (DD and CN) using the ROBINS-I tool, which assesses studies across seven domains: bias due to confounding variables, bias due to the selection of participants, bias in the classification of interventions, bias due to deviations of the intended interventions, bias due to missing data, bias in the measurement of outcomes, and bias in the selection of the reported results. For each domain, the risk of bias can be low, moderate, or serious.

### 2.4. Statistical Analysis

Statistical analyses were performed using Stata v. 17 (StataCorp. 2021 Stata Statistical Software: Release 17, College Station, TX, USA: StataCorp LLC). Risk Ratios (RR) and corresponding 95% confidence intervals (CI) were calculated using a random effects model (DerSimonian-Laird), anticipating high clinical heterogeneity in terms of the patient baseline parameters. Statistical heterogeneity was quantified with the Higgin’s I^2^ statistic; values below 30% represent low heterogeneity, values between 30 and 60% represent moderate heterogeneity, and values above 60% represent substantial heterogeneity. A further subgroup analysis was performed by stratifying the studies by type and continent of origin, and a leave-one-out sensitivity analysis was performed by iteratively removing one study at a time to identify potential outliers influencing the obtained pooled results.

Risk of publication bias was assessed by the visual symmetry of associated funnel plots and using Egger’s and Begg’s tests for every outcome incorporating at least ten studies. For all of the statistical analyses in the present study, a *p*-value below 0.05 was considered to be statistically significant.

## 3. Results

After screening two hundred and ten unique studies and applying the exclusion criteria, thirteen studies, incorporating a total of 6,279,722 patients (750,615 in the BRS group versus 5,529,107 in the control group), were deemed eligible for inclusion in the final quantitative analysis (Figure 1). All of the studies were retrospective cohort studies, six of which utilized propensity score matching to adjust for common confounders. Eight studies originated from North America, while four reported on European patients (Table 1, studies [9,10,12,13,14,15,16,17,18,19,20,21,22]).

### 3.1. Risk for Colorectal Cancer

Based on reports from eleven studies [9,10,12,13,14,15,16,17,18,19,20], patients in the BRS group exhibited a significantly reduced risk for developing colorectal cancer (RR 0.56, 95% CI 0.4–0.8, *p* < 0.001, Figure 2), with substantial interstudy heterogeneity (I^2^ = 95.2%).

When different types of bariatric procedures were considered, sleeve gastrectomy was found to be significantly associated with fewer cases of colorectal cancer (RR 0.55, 95% CI 0.36–0.83, *p* < 0.001, Figure 3A), while gastric bypass and banding did not exert any statistically significant effect (*p* = 0.05 and 0.27, respectively, Figure 3B,C).

The sensitivity analysis did not reveal any single study outliers affecting the obtained cumulative results, solidifying the robustness of the analysis (Figure 4).

### 3.2. Risk of Colorectal Adenomas

The risk of the development of colorectal adenomas was assessed in four studies [15,20,21,22], revealing a non-significant effect of bariatric surgery on their development (RR 0.8, 95% CI 0.54–1.17, *p* = 0.25, Figure 5). Substantial interstudy heterogeneity was encountered in the analysis (I^2^ = 93.4%).

### 3.3. Subgroup Analysis

When the studies were stratified according to origin, the North American studies were found to report a more protective effect of bariatric surgery than the European studies did (Figure 6). Moreover, retrospective propensity matched studies were observed to report a more beneficial effect on the risk of developing colorectal cancer in BRS patients as compared to that of the unmatched studies, albeit with higher interstudy heterogeneity (Figure 7). However, in neither case was a statistically significant intergroup difference encountered during the subgroup analysis (*p* = 0.25).

### 3.4. Publication Bias Assessment

Visual assessment of funnel plots did not reveal any gross asymmetries. Egger’s and Begg’s tests were utilized to quantitatively assess the risk of publication bias in the colorectal cancer incidence outcome (*p* = 0.95 and 0.75, respectively), indicating the absence of publication bias.

### 3.5. Risk of Bias and Critical Appraisal

The data in all of the included studies are derived from databases, of which ten are large cross-sectional databases and three are institutional databases. The average follow-up after bariatric surgery was reported in nine studies. In most cases, the follow-up averaged 60 or more months after bariatric surgery, with a single study reporting a shorter follow-up of 36 months. In the latter case, the reported RR was entirely balanced between the BRS and control group, with a value of one (Figure 2).

Overall, the risk of bias of the included studies was low in three cases, moderate in nine cases, and serious in one study (Table 2). When the components of the ROBINS-I system were examined, the risk of bias was found to be related to confounding variables, patient selection, and missing data. Specifically, four retrospective unmatched studies were, by design, at moderate risk of confounding variables, while the remaining seven propensity score-matched studies were at a low risk. Regarding the risk of selection bias, a serious bias was encountered in a single study that restricted the patient selection to BMI ≥ 40 kg/m^2^ and a moderate one was encountered in six other cases that either lacked information on how the patients were selected or used BMI cut-offs in the range from 30 to 40 kg/m^2^. Finally, a single study was found to be at a moderate risk for missing data bias since the reported average follow-up of 36.4 months might have been inadequate to detect all of the emerging cases of colorectal neoplasias.

## 4. Discussion

In the present systematic review and meta-analysis, bariatric surgery was found to reduce the risk for colorectal cancer by 44%. A trend towards reduced adenoma formation was also observed, albeit to a non-significant degree. Both of these findings suggest a strong protective effect of BRS, which is in line with previously published reports [23,24]. In fact, evidence appears to be accumulating at a consistent pace; since the previous meta-analysis on the topic was published in 2020 [23], five new studies have been published, almost quintupling the total sample size to 6,279,722 patients. Interestingly, as the evidence accrues, the protective effect of BRS becomes more apparent. The meta-analysis by Afshar et al. [24] published in 2014 reported an RR of 0.73, followed-up by an RR of 0.64 by Almazeedi et al. [23]. In this context, the currently calculated RR of 0.56 arguably represents a more accurate reflection of the protective effect exerted by BRS, complementing a trend that has already existed for eight years.

A rational assumption is that the risk of CRC would decrease with bariatric surgery because of the improved metabolic effect after the surgery. Yet, the evidence is, to date, controversial. For example, early bariatric surgery studies did not investigate the CRC risk specifically and showed no change in the CRC risk after bariatric surgery. Recent US and French cohorts that followed the patients for up to 10 years after surgery indicate that bariatric surgery decreases the risk of CRC in female patients [25,26]. Nevertheless, these studies found a smaller or no reduction of CRC risk in male patients. On the contrary, the UK and Nordic studies discovered a puzzling increase in the CRC incidence 5–10 years after bariatric surgery, with a stronger effect on the male population [9,27]. Most of the previous reports though used administrative databases that can lead to a bias if they are not accounted for correctly. Another point that must be highlighted is the fact that Type II diabetes is a component of the metabolic syndrome in morbidly obese patients, which is an independent prognostic factor associated with an increased risk of CRC. A varying degree of metabolic improvement and type II diabetes remission could be the underlying etiology for a differential effect regarding the type of bariatric surgery and CRC risk.

In the present analysis we also explored, within the constraints of the available literature, the role of different bariatric procedures on reducing the risk of CRC. Amongst the evaluated studies, sleeve gastrectomy presented a stronger association with a reduced CRC incidence, followed by gastric bypass and gastric banding (Figure 3). However, two points need to be made in this respect: Firstly, the small number of included studies hints at the possibility of a type II statistical error being present in this sub analysis. Secondly, and perhaps more importantly, the study by Mackenzie et al. [10] was an outlier, contradictorily reporting increased risks for CRC development amongst the patients undergoing gastric bypass. This is an interesting finding that is ostensibly linked to the effect of malabsorptive procedures on the gut microenvironment [10,25,28].

A significant decrease in the incidence rates of colorectal adenomas could not be presently demonstrated, although a trend towards less risk was appreciated. Previous reports indicate that obesity is not only associated with increased risk for adenoma formation [15], but it also exhibits a dose-dependent relationship, with super obese patients being particularly susceptible to adenoma formation [29] and the subsequent conversion to an invasive disease [30]. From this perspective, a heightened index of suspicion for colorectal neoplasias is mandated in obese individuals, and bariatric surgery may be contemplated to impede carcinogenicity [31].

In the complementary subgroup analysis presented herein, geography seemingly plays a central role in the risk colorectal cancer development within the evaluated patient population. North American studies were associated with more protection following BRS (Figure 5). A difference in the overall prevalence of obesity between the USA and Europe may underly this observed geographic variability [26,27], as previous meta-analyses similarly reported higher incidence rates of CRC amongst obese individuals in the USA as opposed to those of their European counterparts [32]. The stratification by study type (matched versus unmatched) did not reveal any significant discrepancies, with the propensity score-matched studies reporting a more modest protective effect of BRS, but with higher interstudy heterogeneity. Further research with large worldwide cohorts and a long follow-up is needed to better determine the risk that different populations carry for the development of CRC after bariatric surgery and how should surveillance be conducted on obese patients depending on whether they have received BRS or not.

While the pooled results obtained from the present analysis indicate a strongly significant protective effect of BRS on CRC development, some important caveats should be acknowledged in addition to those previously discussed. First and foremost, despite the CRC incidence rates amongst the patients in the BRS group being almost halved, an explicit causal relationship between BRS and the observed reduced incidence rates cannot be documented based on the findings of this study. Although the evidence indirectly suggests that weight loss leads to reduction of the risk of CRC [33,34], the success of bariatric procedures in reducing body weight was inconsistently reported in the data extracted from the included studies. Secondly, the length of the follow-up is an important determinant when one is considering CRC emergence. Taube et al. [9], in a recent long-term incidence analysis of the Swedish Obese Subjects registry, indicated that the protective effect of BRS is most evident from 4 to 10 years post-operatively, and it wanes thereafter. A follow-up of five years or less, as was the cases in four of the included studies, may thus be inadequate. In turn, this implies that CRC cases may have been missed due to a short-term follow-up, with unknown implications on the calculated effect size, especially taking into account the contradictory results of the previous Swedish study by Derogar et al. [25], which reported a two-fold increase in the CRC 10 years post-operatively. Additionally, the possibility of selection and confounding biases cannot be entirely ruled out given the nature of the included studies. Finally, while efforts were made to exclude the patient population overlap in the analysis, it cannot be entirely ruled out.

## 5. Conclusions

In conclusion, accumulating evidence points at a significant protective effect of BRS on the development of CRC. In the present analysis, the incidence rate of colorectal cancer was approximately halved the amongst obese individuals that had been operated on. Moreover, sleeve gastrectomy was found to perform better in terms of CRC prevention than gastric bypass does, indicating that malabsorptive procedures have a more modest effect, plausibly due to detrimental changes exerted in the microenvironment of the colon and rectum. Important limitations exist in the present analysis, which could potentially be overcome with a randomized trial. Nevertheless, randomizing obese individuals in surgical and non-surgical cohorts would be especially problematic from an ethical perspective. Ultimately, future case–control studies with long-term follow-up will help to further delineate the exact effect of BRS on the incidence of colorectal neoplasias.

## Figures and Tables

**Figure 1 ijerph-20-03981-f001:**
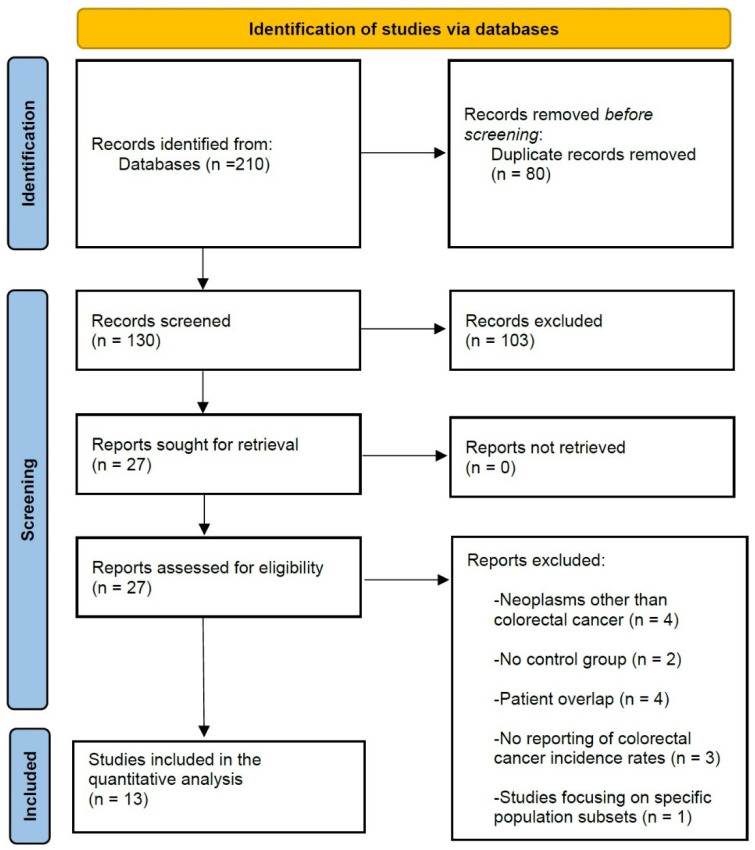
Prisma flowchart of study selection.

**Figure 2 ijerph-20-03981-f002:**
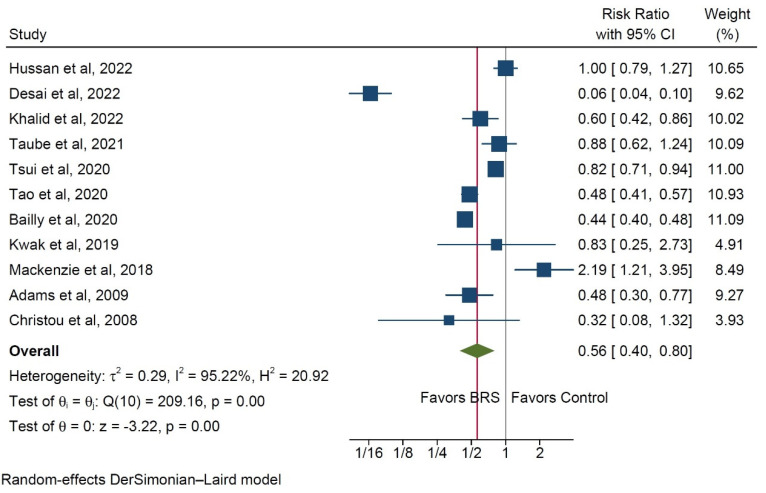
Forest plot of incidence rates of colorectal cancer in the bariatric surgery and control groups [9,10,12,13,14,15,16,17,18,19,20].

**Figure 3 ijerph-20-03981-f003:**
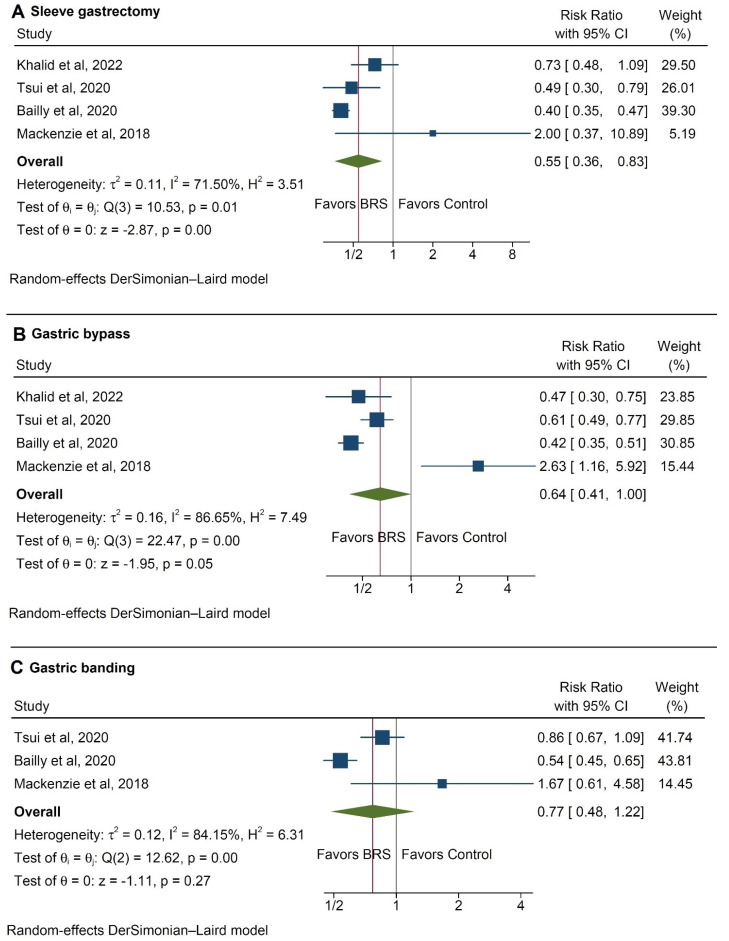
Incidence rates of colorectal cancer between control group and patients with a history of (**A**) sleeve gastrectomy, (**B**) gastric bypass, and (**C**) gastric banding [9,10,12,13,14,15,16,17,18,19,20].

**Figure 4 ijerph-20-03981-f004:**
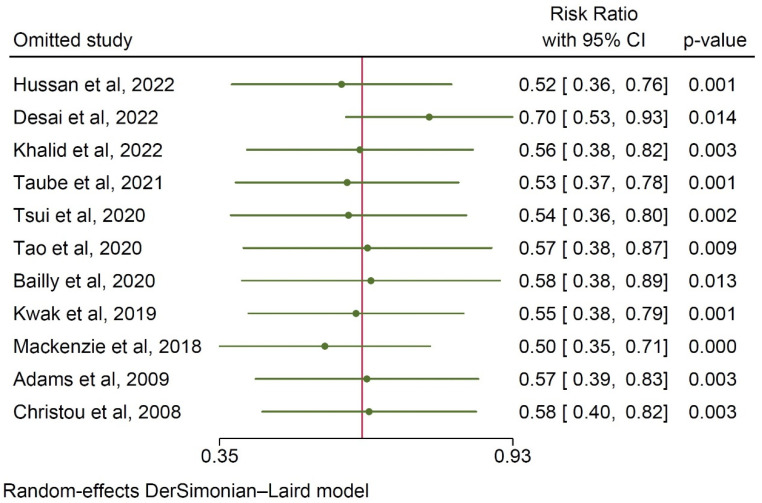
Leave-one-out sensitivity analysis on the incidence rates for colorectal cancer [9,10,12,13,14,15,16,17,18,19,20].

**Figure 5 ijerph-20-03981-f005:**
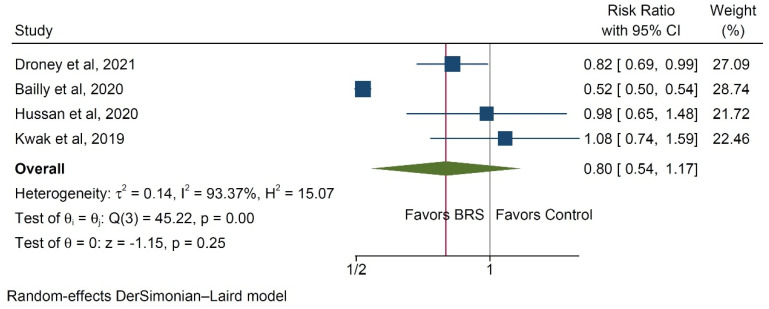
Forest plot of the risk for developing colorectal adenomas [15,20,21,22].

**Figure 6 ijerph-20-03981-f006:**
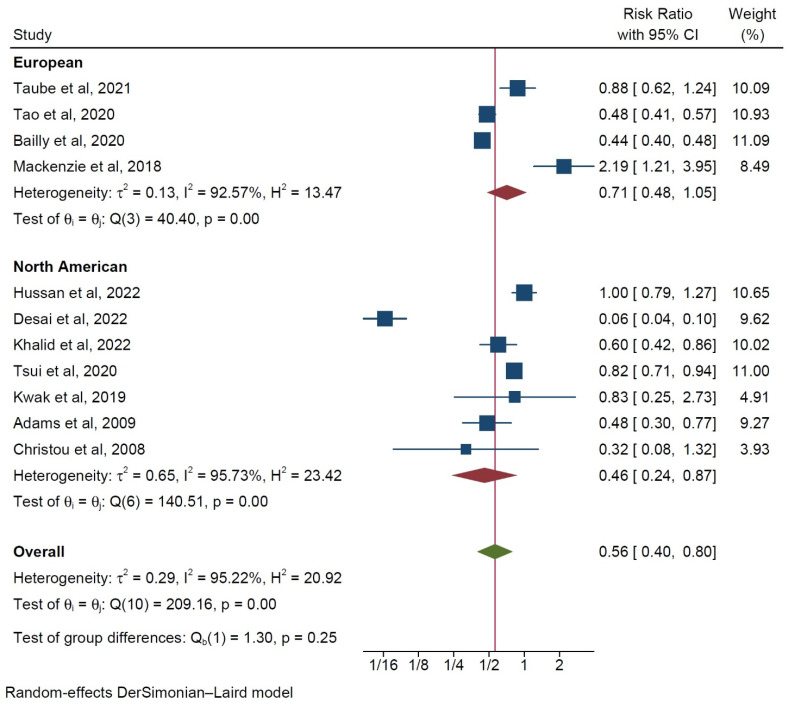
Subgroup analysis of the incidence of colorectal cancer depending on continent of origin [9,10,12,13,14,15,16,17,18,19,20,21,22].

**Figure 7 ijerph-20-03981-f007:**
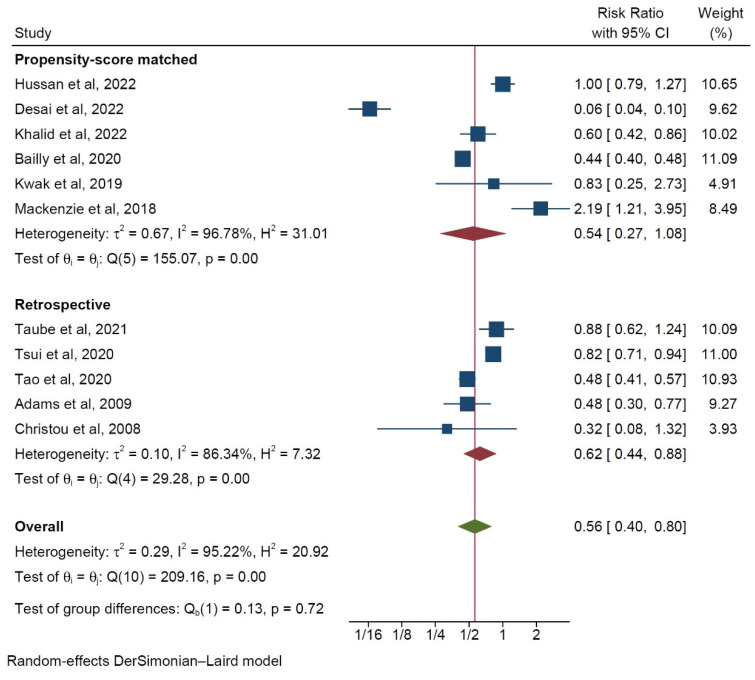
Subgroup analysis of incidence rates of colorectal cancer depending on study type [9,10,12,13,14,15,16,17,18,19,20,21,22].

**Table 1 ijerph-20-03981-t001:** Baseline study and patient demographics.

Author	Study Type	Country of Origin	Interval of Data Collection	Patients, n (%)		Age		Gender (M/F), n (%)	Mean Follow-Up Duration (Months)
				BRS	Control	BRS	Control		
Hussan et al., 2022 [17]	Retrospective, PSM	USA	2012–2020	88,630	327,734	43.4 ± 10.8	43.7 ± 11	19,864 (22.4)/68,766 (77.6)	36.4 ± 24.8
Desai et al., 2022 [13]	Retrospective, PSM	USA	1999–2014	174,765	524,355	43.9 ± 25.2	54.9 ± 36.4	n/a	n/a
Khalid et al., 2022 [19]	Retrospective, PSM	USA	2010–2018	19,272	9636	n/a	n/a	23,742 (82.1)/5166 (17.9)	60
Droney et al., 2021 [21]	Retrospective	USA	2010–2017	417	1360	50.1 ± 10.4	49.9 ± 11.8	516 (29)/1261 (71)	n/a
Taube et al., 2021 [9]	Retrospective	Sweden	1987–2001	2006	2038	47.2 ± 5.9	48.7 ± 6.3	1180 (29.2)/2864 (70.8)	266.4 (172.8) *
Tsui et al., 2020 [14]	Retrospective	USA	2006–2012	71,000	694,500	n/a	n/a	n/a	n/a
Tao et al., 2020 [12]	Retrospective	Multinational	1980–2015	302,576	2,977,526	n/a	n/a	172,039 (31.7)/370,319 (68.3)	n/a
Hussan et al., 2020 [22]	Retrospective, PSM	USA	1994–2018	84	107	55.7 ± 7.8	56.1 ± 5.9	38 (19.9)/153 (80.1)	>60
Bailly et al., 2020 [15]	Retrospective, PSM	France	2009–2018	74,131	971,217	57.3 ± 5.5	63.4 ± 7	n/a	63.6 ± 25.2
Kwak et al., 2019 [20]	Retrospective, PSM	USA	1985–2015	2231	2231	42.6 (10.3) *	42.8 (13.4)	734 (16.4)/3728 (83.6)	93.6 *
Mackenzie et al., 2018 [10]	Retrospective, PSM	England	1997–2012	8794	8794	42 (15)	42 (15)	3450 (19.6)/14,138 (80.4)	55 *
Adams et al., 2009 [16]	Retrospective	USA	1984–2002	6709	9609	38.9 ± 10.3	39.1 ± 10.7	2512 (15.4)/13,806 (84.6)	144 ± 67
Christou et al., 2008 [18]	Retrospective	Canada	1986–2002	1035	5746	45.1 ± 11.6	46.7 ± 13.1	2424 (35.7)/4357 (64.3)	60

BRS = Bariatric Surgery and PSM = Propensity Score Matching. * Represents values as median (interquartile range).

**Table 2 ijerph-20-03981-t002:** Risk of bias summary for non-randomized studies using the ROBINS-I tool. R1 = bias due to confounding, R2 = bias due to selection of participants, R3 = bias in classification of interventions, R4 = bias due to deviations of intended interventions, R5 = bias due to missing data, R6 = bias in measurement of outcomes, and R7 = bias in selection of the reported results.

Study	R1	R2	R3	R4	R5	R6	R7	Overall
Taube et al., 2021 [9]	Moderate	Moderate	Low	Low	Low	Low	Low	Moderate
Mackenzie et al., 2018 [10]	Low	Low	Low	Low	Low	Low	Low	Low
Tao et al., 2020 [12]	Moderate	Low	Low	Low	Low	Low	Low	Moderate
Desai et al., 2022 [13]	Low	Low	Low	Low	Low	Low	Low	Low
Tsui et al., 2020 [14]	Moderate	Low	Low	Low	Low	Low	Low	Moderate
Bailly et al., 2020 [15]	Low	Low	Low	Low	Low	Low	Low	Low
Adams et al., 2009 [16]	Moderate	Moderate	Low	Low	Low	Low	Low	Moderate
Hussan et al., 2022 [17]	Low	Serious	Low	Low	Low	Moderate	Low	Serious
Christou et al., 2008 [18]	Moderate	Low	Low	Low	Low	Low	Low	Moderate
Khalid et al., 2022 [19]	Low	Moderate	Low	Low	Low	Low	Low	Moderate
Kwak et al., 2019 [20]	Low	Moderate	Low	Low	Low	Low	Low	Moderate
Droney et al., 2021 [21]	Moderate	Moderate	Low	Low	Low	Low	Low	Moderate
Hussan et al., 2020 [22]	Low	Moderate	Low	Low	Low	Low	Low	Moderate

## Data Availability

All data are available upon reasonable request.

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
