# Peer review of "The Protective Effect of Bariatric Surgery on the Development of Colorectal Cancer: A Systematic Review and Meta-Analysis"

_ijerph, 2023, doi:10.3390/ijerph20053981_

Round 1

Reviewer 1 Report

Review The protective effect of bariatric surgery on the development of colorectal cancer: a systematic review and meta-analysis

Thank you for the opportunity to read this interesting paper on bariatric surgery and the development of colorectal cancer supporting the idea that bariatric surgery decreases the risk of developing CRC.

Abstract

BRS not explained in abstract, does it make sense to abbreviate bariatric surgery BRS?

Figure 3, the figure would be easier to read if the studied procedure was stated in the headline to the figure ie just next to the letter A, B and C.

The studies by Taube et al and Tao et al have studied the same patients, all patients included by Taube should also be included in Tao’s study. Are there more duplicates among the studies?

There have numerous publications on the topic in the last years:

Almazeedi, S., et al. "Role of bariatric surgery in reducing the risk of colorectal cancer: a meta-analysis." Journal of British Surgery 107.4 (2020): 348-354.

Zhang, Kui, et al. "Effects of bariatric surgery on cancer risk: evidence from meta-analysis." Obesity surgery 30 (2020): 1265-1272.

Wiggins, Tom, Stefan S. Antonowicz, and Sheraz R. Markar. "Cancer risk following bariatric surgery—systematic review and meta-analysis of national population-based cohort studies." Obesity Surgery 29 (2019): 1031-1039.

Please elaborate what news your study brings, apart from including more patients.

Author Response

We thank the reviewer for his/her thorough and careful reading of our manuscript and appreciate the insightful comments which help to improve its quality. Our response follows (the reviewer’s comments are in italics)

1) BRS not explained in abstract, does it make sense to abbreviate bariatric surgery BRS?

Response: We thank the reviewer for pointing this out. We added an explanation for the abbreviation in the abstract.

2) Figure 3, the figure would be easier to read if the studied procedure was stated in the headline to the figure ie just next to the letter A, B and C.

Response: The change was implemented as the reviewer recommended.

3) The studies by Taube et al and Tao et al have studied the same patients, all patients included by Taube should also be included in Tao’s study. Are there more duplicates among the studies?

Response: We thank the reviewer for this very accurate comment. Indeed, the studies by Taube and Tao both include Swedish patients and patient overlap is a possibility. However, these studies derive patient datasets from different databases (the latter from the Nordic Obesity Surgery Cohort and the former from institutional databases). In the present review, we made sure to exclude studies with overlapping patient datasets, originating from the same database. We favored this approach because patient overlap cannot be entirely ruled-in or ruled-out and should, to some degree, be tolerated for the purposes of this analysis. Following the reviewer’s comment we decided to note this in the limitations section of the discussion (lines “260-261”).

4) There have numerous publications on the topic in the last years:

Almazeedi, S., et al. "Role of bariatric surgery in reducing the risk of colorectal cancer: a meta-analysis." Journal of British Surgery 107.4 (2020): 348-354.

Zhang, Kui, et al. "Effects of bariatric surgery on cancer risk: evidence from meta-analysis." Obesity surgery 30 (2020): 1265-1272.

Wiggins, Tom, Stefan S. Antonowicz, and Sheraz R. Markar. "Cancer risk following bariatric surgery—systematic review and meta-analysis of national population-based cohort studies." Obesity Surgery 29 (2019): 1031-1039.

Please elaborate what news your study brings, apart from including more patients.

Response: We thank the reviewer for the question and the opportunity to answer. We do recognize that similar meta-analyses were previously published and in fact cite them in-text (“lines 182-183”). Apart from including more patients and conducting a separate complementary analysis for colorectal adenomas, the present meta-analysis employs subgroup analyses to investigate the impact of geography and type of surgery on the observed protective effect of bariatric surgery. Finally, as stated in-text (“lines 189-191”), the present “updated” meta-analysis reveals that the magnitude of the observed protective effect may be even greater, as evidence continues to accrue.

Reviewer 2 Report

This systematic review addresses a relevant topic with important implications for patients’ health. The research is overall well performed and the results are sound, despite most of source studies presented at least a moderate risk of bias.

Some suggestions to improve the presentation of the results are provided below:

Methods:

1. Although not compulsory, please indicate if this systematic review was registered at any databases (such as PROSPERO or INPLASY)

2. Please confirm in your inclusion criteria if documents written in any language were considered.

3. According to PRISMA statement, the complete search strategy should be provided for at least one the databases screened for documents.

4. Table 1. Although it appears detailed in the text, the meaning of the abbreviation BRS should be detailed in a table footer so that the table is self-explanatory without the need to consult the text.

Author Response

We thank the reviewer for his/her time and insightful comments that help strengthen our manuscript. Our response follows (the reviewer’s comments are in italics)

1) Although not compulsory, please indicate if this systematic review was registered at any databases (such as PROSPERO or INPLASY)

Response: We thank the reviewer for the question. The systematic review was indeed registered on PROSPERO with the ID: CRD42023387393. We went ahead and added this information in-text.

2) Please confirm in your inclusion criteria if documents written in any language were considered.

Response: We thank the reviewer for the recommendation. All language were considered; this is now added in-text (“line 63”).

3) According to PRISMA statement, the complete search strategy should be provided for at least one the databases screened for documents.

Response: We thank the reviewer for pointing this out. The search for Medline database was added in-text (“lines 57-58”).

4) Table 1. Although it appears detailed in the text, the meaning of the abbreviation BRS should be detailed in a table footer so that the table is self-explanatory without the need to consult the text.

Response: We thank the reviewer for the suggestion. An explanatory footer was added in Table 1.